# *Zingiber officinale* Extract (ZOE) Incorporated with Layered Double Hydroxide Hybrid through Reconstruction to Preserve Antioxidant Activity of ZOE against Ultrasound and Microwave Irradiation

**DOI:** 10.3390/nano9091281

**Published:** 2019-09-08

**Authors:** Hyoung-Jun Kim, Su-Bin Lee, Ae-Jin Choi, Jae-Min Oh

**Affiliations:** 1Department of Energy and Materials Engineering, Dongguk University-Seoul, Seoul 04620, Korea; 2Department of Agrofood Resources, National Institute of Agricultural Sciences of RDA, Wanju 55365, Korea

**Keywords:** layered double hydroxide, *Zingiber officinale* extract, antioxidant activity, ultrasound, microwave irradiation, protection

## Abstract

We prepared *Zingiber officinale* extract (ZOE) incorporated in a layered double hydroxide (LDH) hybrid through a reconstruction method in order to preserve the antioxidant activity of ZOE from ultrasound and microwave irradiation. X-ray patterns, infrared spectroscopy, and scanning electron microscopy suggested that ZOE moieties were encapsulated in the interparticle space of reconstructed LDH, thus preserving its intact structure. Dynamic light scattering and zeta-potential measurement also supported the hypothesis that ZOE moieties were located in the interparticle pore of LDH rather than at the surface of LDH particles. Thermogravimetry analysis revealed that thermal stability of encapsulated ZOE could be enhanced by LDH encapsulation. Radical scavenging assay showed that antioxidant activity of ZOE–LDH hybrid was increased after ultrasound and microwave irradiation, while ZOE itself dramatically lost its antioxidant activity upon ultrasound and microwave treatment.

## 1. Introduction

Many natural plant extracts have antioxidant activities due to their bioactive components, such as flavonoids, phenolics, and others [1,2,3,4]. *Zingiber officinale* (ZO), generally known as ginger, belongs to the family Zingiberaceae, and it has been recently studied due to its nutraceutical effects like preventing oxidative stress in an animal model [5]. Gargova et al. have reported that supercritical CO_2_ extraction of ginger extract possesses good radical scavenging activity due to its high polyphenol content [6]. On the other hand, essential oil of ginger is known to have high antioxidant properties. It is often used as a functional food additive [7]. The antioxidant activity of ZO is mainly attributed to its phytochemicals, such as 6-gingerol, 8-gingerol, and 6-shogaol [8]. However, these phytochemicals can be destroyed by external damage such as ultrasound and microwave irradiation [9]. Irradiation with ultrasound can cause cleavage of the H–O bond of H_2_O in an aqueous solution, resulting in hydroxyl radical and hydrogen atoms [10].

These radicals can react with organic compounds, leading to degradation of phytochemicals [11]. Sun et al. have reported that degradation of pelargonidin 3-glucoside extracted from strawberry upon ultrasound irradiation can reduce its antioxidant capacity depending on the ultrasound power [12]. Microwave irradiation can induce rapid and intensive heating of polar substances, thus accelerating the reactivity of molecules. These phenomena can decompose phytochemicals in natural plant extracts. Concentrations of bioactive compounds, such as chlorophyll, vitamin C, polyphenol, and flavonoid in Brussels sprouts, and radical scavenging activity have been reported to be significantly reduced with increasing microwave treatment time [13]. The antioxidant activity of ginger extract can also be significantly reduced upon microwave irradiation [14]. Therefore, external physiochemical stimuli such as ultrasound and microwave irradiation can negatively influence the antioxidant activity of phytochemicals.

To overcome this problem, many researchers have used reservoir materials, such as lipid, polysaccharide, protein, and inorganic nanoparticles to preserve biologically active but physiochemically fragile moieties [15,16,17]. Among these reservoirs, layered double hydroxide (LDH) is one of the emerging materials in terms of encapsulation, protection, and controlled release of biologically active species, such as drugs, therapeutic genes, vitamins, and others [18,19,20]. LDH structures have a positively charged metal hydroxide layer and charge-compensating interlayer anions. They usually have a relatively large specific surface area and high anion exchange capacity [21]. LDHs can encapsulate a large amount of anion species protected by LDH layers in terms of thermal, mechanical, chemical, and biological stimulations [22,23]. In our previous study, we have reported that DNA intercalation using LDH by ion-exchange can protect DNA from an external attack of DNA-destroying enzymes [24]. Preservation of vitamin C, which is vulnerable to oxidation under ambient conditions, into the interlayer of LDH can enhance its chemical stability [25]. Furthermore, it has been reported that LDH could effectively encapsulate natural extract that contains a variety of sized molecules by reconstruction method. Sand-rose or house-of-cards structure obtained by reconstruction of LDH can provide the interparticle pores to load extract moiety with various molecular weights [26].

The objective of this study was to hybridize *Zingiber officinale* extract (ZOE) and LDH using a reconstruction method and to investigate its antioxidant activity protection effect. Structures of pristine MgFe–CO_3_–LDH, calcined LDH, and ZOE-incorporated LDH (ZOE–LDH) hybrid were analyzed using powder X-ray diffractometer and Fourier transform infrared spectroscopy. Morphologies, particle sizes, and surface charges of pristine MgFe–CO_3_–LDH and ZOE–LDH hybrid were measured by scanning electron microscopy, dynamic light scattering, and zeta-potential, respectively. To determine the chemical formula of a ZOE–LDH hybrid and the thermal stability of ZOE, a thermogravimetry analysis was carried out. Finally, preservation of the antioxidant activity of ZOE by LDH encapsulation was valuated utilizing radical scavenging assay after appropriate treatment with ultrasound or microwave irradiation.

## 2. Materials and Methods

### 2.1. Materials

Magnesium nitrate hexahydrate (Mg(NO_3_)_2_∙6H_2_O) was obtained from Junsei Chemical CO., LTD (Tokyo, Japan). Iron nitrate nonahydrate (Fe(NO_3_)_3_∙9H_2_O) and 2,2-diphenyl-1-picrylhydrazyl (DPPH) were purchased from Sigma-Aldrich, Co. Inc. (St. Louis, MO, USA). Sodium nitrate (NaOH), sodium bicarbonate (NaHCO_3_), and methyl alcohol, 99.5% were acquired from DAEJUNG CHEMICALS & MATERIALS CO., LTD (Siheung, Korea). Dimethyl sulfoxide (DMSO) was obtained from TOKYO CHEMICAL INDUSTRY CO., LTD (Tokyo, Japan). *Zingiber officinale* extract (ZOE) using enzyme extraction method was provided by the Rural Development Administration, Republic of Korea.

### 2.2. Preparation of Pristine MgFe–CO_3_–LDH

Pristine MgFe–CO_3_–LDH was synthesized by conventional co-precipitation method. A solution containing Mg(NO_3_)_2_∙6H_2_O (0.05 M) and Fe(NO_3_)_3_∙9H_2_O (0.025 M) was titrated with NaOH/NaHCO_3_ (0.25 M/0.075 M) solution until the pH reached 9.5. It was then aged for 24 h. The obtained precipitate was centrifuged, washed with deionized water (DW), and then lyophilized.

### 2.3. Preparation of ZOE–LDH Hybrid

To encapsulate ZOE by LDH, pristine MgFe-CO_3_-LDH was first calcined at 400 °C for 9 h in a muffle furnace to obtain MgFe-layered double oxide (MgFe–LDO). ZOE was then incorporated with LDH to prepare hybrid ZOE–LDH by reconstruction route. Typically, 0.741 g of ZOE was dissolved in 100 mL of 10% dimethyl sulfoxide (DMSO) solution and then 1.742 g of MgFe–LDO powder was dispersed in ZOE solution. The suspension was stirred for 24 h under N_2_ gas condition. A powder, as the final product, was obtained by lyophilization of centrifuged and washed precipitate.

### 2.4. Characterization

Powder X-ray diffraction (PXRD) patterns of ZOE, MgFe–LDH, MgFe–LDO, and ZOE–LDH hybrid were investigated using Ultima IV (Rigaku, Tokyo, Japan) with Cu K_α_ radiation (λ = 1.5406 Å). Diffraction patterns were obtained in the 2θ range from 5 to 80° with a scanning rate of 5°/min. The intact structure of ZOE in the ZOE–LDH hybrid was identified by Fourier transform infrared spectroscopy (FT-IR, Frontier MIR/FIR spectrometer, Perkin Elmer, Waltham, MA, USA) with a scanning range from 4000 to 400 cm^−1^ using the conventional KBr method. Particle morphologies of ZOE and its hybrid were examined by field emission-scanning electron microscopy (FE-SEM, JSM-7100F, JEOL-USA Inc., Peabody, MA, USA)). The hydrodynamic radius and surface charge of pristine LDH and ZOE–LDH hybrid were investigated by dynamic light scattering (DLS) and zeta-potential using an ELSZ-1000 analyzer (Otsuka, Kyoto, Japan). For zeta-potential and DLS, 1 mg of each powder was dispersed in 10 mL of DW. The content of ZOE in ZOE–LDH hybrid was evaluated by measuring the mass of the initial and remnant of ZOE in the reaction vessel. Thermogravimetry analysis (TG) (SDT Q600, TA Instruments, New Castle, DE, USA) of ZOE and ZOE–LDH hybrid was carried out with a heating rate of 10 °C/min under air gas condition from 30 to 1000 °C. The chemical formula of ZOE–LDH hybrid was determined based on the ZOE content and the TG result of the ZOE–LDH hybrid.

### 2.5. Radical Scavenging Activity

To evaluate the ability of LDH in preserving the antioxidant activity of the payload, DPPH assay was carried out for both ZOE and the ZOE–LDH hybrid before and after treatment with ultrasound and microwave irradiation. ZOE solution and ZOE–LDH suspension were prepared in 10% DMSO to obtain a concentration of 1 mg/mL. The prepared solution and suspension were then exposed to harsh conditions using microwave for 0, 1, and 3 min, and ultrasound for 0, 5, and 30 min, respectively. Both samples were then diluted to have ZOE concentrations of 1000, 500, 250, 125, 62.5, 31.25, 15.623, 7.8123, and 3.906 ppm. For comparison, MgFe–LDH suspension was treated with ultrasound or microwave irradiation and diluted so that the concentration of the inorganic part was the same as that of the ZOE–LDH suspension. For DPPH assay, 200 µL of each sample was mixed with 800 µL of DPPH solution (3.8 × 10^−4^ M of DPPH in 80% MeOH). As a negative control, 200 µL of 80% MeOH without sample was mixed with 800 µL of DPPH solution. A sample blank was prepared by replacing DPPH solution with 80% MeOH for each mixture. The mixture was agitated for 30 min at room temperature under dark conditions. The absorbance of each sample was then measured at wavelength of 517 nm. The antioxidant activity was calculated using the equation below:Antioxidant activity (%) = [control absorbance − (sample absorbance − blank absorbance)/control absorbance] × 100

## 3. Results and Discussion

To investigate the crystal structures of ZOE, MgFe–LDH, MgFe–LDO, and ZOE–LDH hybrid, X-ray diffraction (XRD) patterns of powdered samples were obtained. The XRD pattern of MgFe–CO_3_–LDH showed characteristic peaks of pyroaurite (JCPDS. No 14-0293) at 11.04, 23.15, 33.73, 37.86, 59.43, and 60.81° for (003), (006), (012), (015), (110), and (113), respectively as shown in Figure 1b [27]. After calcination, pyroaurite phase of MgFe–CO_3_–LDH transformed to periclase (MgO; JCPDS. No. 45-0946) with peak position at 35.36, 42.68, and 62.11° corresponding to (111), (200), and (220) reflections, respectively, as shown in Figure 1c [28]. ZOE itself was found to be amorphous, exhibiting a broad pattern between 15 and 20°, possibly due to random assembly among carbohydrate residues in ZOE moiety as shown in Figure 1a. In the XRD pattern of the ZOE–LDH hybrid, (003), (012), (110), and (113) peaks corresponding to pyroaurite were observed, indicating recovery of the original LDH phase upon reconstruction as shown in Figure 1d. The peak position of (003) in ZOE–LDH hybrid was the same as that in pristine LDH, suggesting that the relatively large molecule in ZOE was not incorporated into the interlayer space of LDH. After hybridization, crystallite size along (003) and (110) plane of LDH decreased from 7.1 and 11.3 nm to 4.4 and 6.1 nm, respectively (calculated by Scherrer’s equation) [29]. The reduction of crystallinity was attributed to partial disorder and reorganization of the LDH lattice during the reconstruction process. Similar phenomena have been reported in the reconstruction of LDH [30]. Periclase peak of LDO was observed in the ZOE–LDH hybrid, showing crystallite size (along (200) plane of periclase) reduction from 7.0 to 5.4 nm. Why LDO did not fully recover the LDH structure was unclear. Large biomolecules in ZOE might have disturbed the hydration of MgO to MgFe–LDH. Reduction in the crystalline size of LDO after reconstruction suggested that the phase transformation from LDO to LDH was hindered during the process. This might be due to the action of various organic moiety in ZOE. Although we could not obtain pure LDH phase in the ZOE–LDH hybrid, the existence of the LDO phase was not disadvantageous in this study. Periclase is known to have several advantages in biological and environmental applications, such as limited solubility in water, inherent biocompatibility, environmental friendliness, high melting point, and low heat capacity [31,32]. Thus, the existence of periclase in the hybrid can enhance the protection ability of inorganic shell along with the physicochemical stability of LDH [33,34].

To identify incorporated moieties in the ZOE–LDH hybrid, FT-IR analyses of ZOE, pristine MgFe–LDH, and ZOE–LDH were carried out. The FT-IR spectrum of MgFe–LDH showed a characteristic stretching peak of M–O for LDH at 575.6 cm^−1^, as shown in Figure 2a [35,36]. In the IR spectrum of ZOE, C–O–C stretching of dialkyl ether, C–O–C stretching of lignin, and C–C stretching of cellulose were observed at 1152, 1079, and 1024 cm^−1^, respectively, as shown in Figure 2c [37,38]. After hybridization, M–O stretching of LDH and C=O, C–O–C, and C–O stretching of ZOE co-existed in the IR spectrum of the ZOE–LDH hybrid, as shown in Figure 2b. These results demonstrated that the intact structure of ZOE moieties was preserved safely in ZOE–LDH hybrid after the reconstruction reaction.

The morphology of ZOE–LDH hybrid was investigated using SEM. The image in Figure 3a shows the large and glassy particles of ZOE. Taking into account the amorphous nature of ZOE as shown in Figure 1a, the glassy morphology of ZOE might be attributed to the intermolecular interaction among organic moiety in the natural extract. In the SEM image of the ZOE–LDH hybrid, two kinds of characteristic morphology, house-of-card and coin-like shape, both with agglomerated phase of 79.0 ± 8.84 nm-sized primary particles, were observed as shown in Figure 3c. They might be attributed to reconstructed LDH [39] and LDO [40], respectively, as shown in Figure 3b. The house-of-card morphology of reconstructed LDH was attributed to the increase of face-to-edge interaction when LDH underwent disorder and reorientation during the reconstruction process. The house-of-card structure can provide extract moiety with interparticle space for protective encapsulation. It was worthy to note that glassy-surface morphology was not observed in the SEM image of the ZOE–LDH hybrid, suggesting that the intermolecular interaction in ZOE was successfully prevented by LDH/LDO moiety and that the ZOE was safely incorporated in the inorganic lattice.

To confirm the location of the organic moiety of ZOE in the hybrid, surface charges of pristine LDH, ZOE, and ZOE–LDH hybrid were examined by measuring their zeta-potentials. As reported previously, the zeta-potential value of pristine MgFe–LDH was fairly positive at 13.30 ± 1.85 mV (pH 6.97), as shown in Figure 4 as a solid line [41]. On the other hand, ZOE had a negative zeta-potential of −24.12 ± 1.09 mV (pH 7.01), possibly due to abundant δ-moiety and anionic sites carbohydrate, lipid, polyphenol, and others, as shown in Figure 4 as a dashed line [42]. After encapsulation of ZOE with LDH, the zeta-potential was found to be 4.53 ± 0.76 mV (pH 7.06), as shown in Figure 4 as open circles, which was between zeta-potential values of ZOE and LDH but leaned towards that of LDH. If ZOE exhibited an outer surface of LDH after hybridization, the zeta-potential of ZOE–LDH would be more biased towards ZOE. As zeta-potential of ZOE–LDH still resided in the positive region, it suggested that ZOE was encapsulated in the interparticle space of LDH, rather than attached on the surface of LDH. This result was similar to our previous result that *Angelica gigas* Nakai extract incorporated in LDH hybrid showed positive zeta-potential [43]. Based on XRD patterns, IR spectra, SEM images, and zeta-potential, we could conclude that MgFe–LDH lost its lamellar structure upon calcination, and the original layered structure was partially recovered to develop the interparticle space of LDH with intact ZOE moiety.

In this regard, we expect that ZOE in the ZOE–LDH hybrid can be well protected in the interparticle cavity of LDH from external harsh conditions and that its biological function, such as antioxidant properties could be preserved in the hybrid, as shown in Scheme 1.

The TG curves and corresponding first derivatives of ZOE and the ZOE–LDH hybrid are shown in Figure 5. Temperature-dependent weight loss of ZOE occurred in the wide temperature range between 25 and 1000 °C with a corresponding weight loss of 88.32%, as shown in Figure 5(Aa). Detailed analyses with the first derivative of the TG curve revealed that this range consisted of steps at 154.6, 195.4, and 293.8 °C as shown in Figure 5(Ba). ZOE–LDH showed two-step weight loss from 25 to 154.2 °C and 154.2 to 647.5 °C with corresponding weight loss of 10.0% and 15.40%, as shown in Figure 5(Ab). The first step was attributed to the dehydration of surface or interlayer absorbed water [44], while the second step could be due to thermal decomposition of the encapsulated ZOE moiety and dehydroxylation of the LDH lattice together [45]. Similar to thermal decomposition of ZOE, the ZOE–LDH hybrid showed three steps of decomposition at 226.3, 338.7, and 436.3 °C, as shown in Figure 5(Bb). Although it was not clear whether all three steps were attributed to the thermal decomposition of ZOE moiety, it was worthy to note that the decomposition temperature points were higher than those of the major decomposition of ZOE alone as shown when comparing Figure 5(Ba) with Figure 5(Bb). We could, therefore, conclude that ZOE in the hybrid was protected by the LDH/LDO particle to acquire thermal stability. TG data and weight differences in extract quantification suggested that the chemical formula of the ZOE–LDH hybrid was (Mg_2_FeO_3.5_) [Mg_2_Fe(OH)_6_(CO_3_)_0.5_]_0.77_·(ZOE)·2.34H_2_O, where ZOE content was 9.68 wt/wt%.

To evaluate the antioxidant activities of the ZOE and ZOE–LDH hybrid, a radical scavenging assay was carried out before and after ultrasound and microwave irradiation on ZOE and ZOE–LDH. The concentration-dependent inhibition curve of ZOE shifted to the right side after ultrasound and microwave irradiation, as shown in Figure 6A, meaning reduced radical scavenging activity of ZOE. On the other hand, the inhibition curve of ZOE–LDH showed enhanced antioxidant activity, as shown in Figure 6B,C. The enhancement of antioxidant activity of ZOE–LDH after ultrasound and microwave irradiation occurs only in high ZOE concentration. The inhibition curve of ZOE–LDH resembles the Hill–Langmuir equation which usually reflects ligand binding to macromolecules in biochemistry and pharmacology. In the Hill–Langmuir equation, K_A_ indicates the standard concentration for representing activity [46]. K_A_ values of ultrasound (5 min) and microwave (1 min)-treated ZOE–LDH were 86.36 and 84.96 ppm, respectively. This result revealed ZOE–LDH did not show antioxidant activity at a concentration lower than 86.36 ppm, therefore, there was no increasing antioxidant activity at a lower ZOE concentration. On the other hand, since antioxidant activity begins to appear at high ZOE concentration, their antioxidant activity increased through the protecting effect by inorganic LDH and LDO. Detailed results of antioxidant activities of ZOE and the ZOE–LDH hybrid are shown in Table 1. IC_50_ values (the concentration needed to inhibit 50% of antioxidant activity) of ZOE and the ZOE–LDH hybrid for radical scavenging activity were determined to be 201.81 ± 13.59 and 349.17 ± 47.50 ppm, respectively. As ZOE concentration in ZOE itself and ZOE–LDH was the same, the maximum IC_50_ value of the ZOE–LDH hybrid must be 201.81 ± 13.59 ppm. However, the ZOE–LDH hybrid showed an increased IC_50_ value, indicating reduced antioxidant activity. The lower antioxidant activity of the ZOE–LDH hybrid was due to captured ZOE moieties in the interparticle pores of LDH and LDO, which could not interact with the DPPH radical. After ultrasound and microwave irradiation, ZOE presented increased IC_50_ value as 1003.29 ± 79.82ppm for ultrasound irradiation and 353.73 ± 29.93 ppm for microwave irradiation, revealing decreased antioxidant activity. On the contrary, radical scavenging activity of ZOE–LDH was increased upon ultrasound and microwave irradiation, showing decreased IC_50_ values as 270.65 ± 28.78 ppm (for 5 min) and 259.36 ± 25.42 ppm (for 1 min), respectively. The enhanced radical scavenging activity might be due to the release of ZOE moieties in the interparticle cavity of LDH and LDO after ultrasound and microwave irradiation. Although the antioxidant activity of ZOE–LDH after ultrasound and microwave irradiation was enhanced, the IC_50_ value was still higher than the IC_50_ value of non-treated ZOE (maximum radical scavenging activity). This is due to partially remained ZOE in the interparticle cavity of LDH and LDO. Moreover, upon increasing irradiation time, the IC_50_ values of ZOE–LDH revealed 252.46 ± 12.47 ppm for ultrasound (30 min) and 231.82 ± 19.00 ppm for microwave (3 min), which was the same values with shorter irradiation time as the 99% confidence intervals calculated by the Student’s *t*-test. This result was attributed to insulation, radical blocking, and microwave absorption properties of LDH and LDO. Thus, incorporated ZOE moieties were protected when temperature was increased and OH radical was formed by ultrasound and microwave irradiation. From these results, we concluded that ZOE was effectively protected by LDH and LDO, thus having suitable physicochemical properties.

## 4. Conclusions

We prepared ZOE-incorporated LDH using a reconstruction process to protect ZOE moieties from harsh conditions. XRD patterns and FT-IR spectra data confirmed that ZOE moieties were incorporated into LDH and LDO due to partial reconstruction of LDO, which has insulation and microwave absorption properties. According to SEM, zeta-potential, and dynamic light scattering analyses, ZOE was incorporated into the interparticle pores of LDH and LDO formed by a few particles. TGA/DTG analysis and measurement of ZOE content revealed that 9.68% of ZOE moieties in the ZOE–LDH hybrid were protected by LDH and LDO. DPPH assay results indicated that antioxidant activity of ZOE was decreased after ultrasound and microwave irradiation. On the other hand, the radical scavenging activity of ZOE–LDH was increased due to released ZOE moieties which protected ZOE from OH radical and high temperature because ZOE moieties were found in the interparticle pores of LDH and LDO during ultrasound and microwave irradiation. These results suggest that the antioxidant activity of ZOE can be effectively preserved by hybridization with LDH using a reconstruction process.

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
