# Peer review of "Zingiber officinale Extract (ZOE) Incorporated with Layered Double Hydroxide Hybrid through Reconstruction to Preserve Antioxidant Activity of ZOE against Ultrasound and Microwave Irradiation"

_nanomaterials, 2019, doi:10.3390/nano9091281_

Round 1
Reviewer 1 Report
This work deals with the preparation of a Zingiber officinale extract (ZO) incorporated layered double hydroxide (LDH) hybrid through reconstruction method in order to preserve its anti-oxidant activity from ultrasound and microwave irradiation. The structure of the manuscript and experiments are well defined and exposed. The study appears to have been carefully conducted however, it deals with a topic that, in my opinion, is not related to the journal aim since no nanomaterial is produced. I the editor accept this premise I suggest the publication after minor revision. The SEM discussion and images should be improved to better understand the showed structures. There is no clear information about the dimensions of the produced structures. Please further discuss Figure 6 about the increase in inhibition percentage at high extract concentration for ZO-LDH.
Author Response
Point 1: This work deals with the preparation of a Zingiber officinale extract (ZOE) incorporated layered double hydroxide (LDH) hybrid through reconstruction method in order to preserve its anti-oxidant activity from ultrasound and microwave irradiation. The structure of the manuscript and experiments are well defined and exposed. The study appears to have been carefully conducted however, it deals with a topic that, in my opinion, is not related to the journal aim since no nanomaterial is produced. I the editor accept this premise I suggest the publication after minor revision. The SEM discussion and images should be improved to better understand the showed structures. There is no clear information about the dimensions of the produced structures. Please further discuss Figure 6 about the increase in inhibition percentage at high extract concentration for ZOE-LDH.
Response 1: Thanks for reviewer’s kind comments. Based on the reviewer’s advice, we measured the primary particle size of ZOE-LDH hybrid. The average particle size of ZOE-LDH hybrid was 79.0 ± 8.84 nm. Although particle formed agglomerates, the primary particle of ZOE-LDH was nanoparticles. Therefore, in terms of primary particle size, ZOE-LDH is nanomaterial. We added more magnified SEM images of ZOE-LDH as Figure 3 (c) and discussion as below.
Figure 3. Scanning electron microscope image of (a) ZOE, (b) ZOE-LDH hybrid and (c) magnified ZOE-LDH hybrid
Page 5, line 176; with agglomerated phase of 79.0 ± 8.84 nm sized both primary particles, were observed (Figure 3 (c)).
Response 2: As reviewer pointed out, in radical scavenging result of ZOE-LDH show enhancement at high concentration. The inhibition curves of ZOE-LDH resemble the Hill-Langmuir equation plot which reflects ligand binding to macromolecules in biochemistry and pharmacology. This equation also explains activity of drugs depending on drug concentration. In hill equation, KA is constant related ligand occupying half of the binding sites showing standard concentration for appearing activity. Thus, we calculated the KA values of ultrasound and microwave irradiated ZOE-LDH hybrid. KA value of ultrasound and microwave irradiated ZOE-LDH was 86.36 ppm and 84.96 ppm, respectively. This result indicated that anti-oxidant activity appear above 84.96 ppm. Therefore, in lower ZOE concentration than 84.96 ppm, there was no meaningful activity. And at higher ZOE concentration than 84.96 ppm, anti-oxidant activity enhanced depending on increasing ZOE concentration through protecting effect of ZOE by inorganic LDH and LDO. We added corresponding discussion in result & discussion section as below.
Page 8, line 243 ~ 252; The enhancement of anti-oxidant activity of ZOE-LDH after ultrasound and microwave irradiation occurs only in high ZOE concentration. Inhibition curve of ZOE-LDH resembles with Hill-Langmuir equation which usually reflects ligand binding to macromolecules at biochemistry and pharmacology. In Hill-Langmuir equation, KA indicated standard concentration for representing activity. [46] KA values of ultrasound (5 mins) and microwave (1 min) treated ZOE-LDH was 86.36 ppm and 84.96 ppm, respectively. This result revealed ZOE-LDH did not show anti-oxidant activity at lower concentration than 86.36 ppm, therefore, there was no increasing anti-oxidant activity at lower ZOE concentration. On the other hand, since anti-oxidant activity begins to appear at high ZOE concentration, their anti-oxidant activity increased through protecting effect by inorganic LDH and LDO.

Reviewer 2 Report
This manuscript by Kim et al. reports on synthesis of Zingiber officianle extract (ZO) housed in a nanocage formed in layered double hydroxide (LDH) and its anti-oxidant activity. The hybrid structure served to enhance the stability of ZO under high temperature. On the other hand, it also led to reduced anti-oxidant activity of ZO, which was explained as a result of the LDH structure acting to hinder the inner ZO moieties to interact with DPPH radicals. Nonetheless, the LDH-protection enabled to even enhance the anti-oxidant activity after ultrasound and microwave irradiation, which is in sharp contrast to non-protected ZO that usually degrades the activity.
This is a well-written paper providing a novel synthetic strategy to preserve the anti-oxidant activity of ZO exposed to ultrasound and microwave irradiation. I recommend publication after the authors addressed the following points:
1. It is not straightforward to conclude that ZO-LDH can provide better anti-oxidant activity from Figure 6. This is because of the fact that the microwave-treated samples, either ZO or ZO-LDH, showed comparable activity, which implies that under different irradiation conditions, e.g. shorter sonication time/power, ZO can have better activity than ZO-LDH. The authors should therefore show how the activity changes by the sonication conditions to support their conclusion.
2. Related to the above point, it is interesting to see whether ZO-LDH activity may be degraded under longer ultrasonication time if ZO moieties are really released after the irradiation as claimed by the authors.
3. The influence of sonication depends on the power of sonicator used. The author should give the power of their sonicator along with a reason why they employed it. It is also not clear why the irradiation time was much longer for the ultrasonication than the microwave irradiation.
4. The abbreviation ZO should be defined when first appear in the main text.
Author Response
Point 1: It is not straightforward to conclude that ZO-LDH can provide better anti-oxidant activity from Figure 6. This is because of the fact that the microwave-treated samples, either ZO or ZO-LDH, showed comparable activity, which implies that under different irradiation conditions, e.g. shorter sonication time/power, ZO can have better activity than ZO-LDH. The authors should therefore show how the activity changes by the sonication conditions to support their conclusion.
Response 1: In Figure 6, anti-oxidant activity of ZO extract itself was reduced after ultrasound and microwave irradiation. On the other hand, anti-oxidant ZO-LDH hybrid was enhanced after ultrasound and microwave irradiation. Moreover, in shorter irradiation time, ZO-LDH shows similar anti-oxidant activity compared with longer irradiation condition. In addition, microwave treated ZO and ZO-LDH indicate different anti-oxidant activity. ZO-LDH hybrid (259.36 ppm of IC50) have higher anti-oxidant activity compared with ZO (353.73 ppm of IC50). Therefore, we think that ZO-LDH hybrid system was better than ZO itself in terms of ZO activity protection from harsh condition.
Point 2: Related to the above point, it is interesting to see whether ZO-LDH activity may be degraded under longer ultrasonication time if ZO moieties are really released after the irradiation as claimed by the authors.
Response 2: Thanks for reviewer’s valuable comment. We carried out DPPH assay of ultrasound and microwave treated ZO and ZO-LDH during two kinds of time condition. For ultrasound irradiation, we treated ultrasound to ZO-LDH during 5 mins and 30 mins. Microwave irradiated for 1 min and 3 mins to ZO-LDH hybrid. The IC50 value of ultrasound irradiated ZO-LDH for 5 mins and 30 mins was 270.65 ± 28.78 ppm and 252.45 ± 12.47 ppm, respectively. After microwave irradiation, IC50 value was 259.36 ± 25.42 ppm for 1 min irradiation and 231.82 ± 19.00 ppm for 3 mins irradiation. Through Student’s t-test, we confirmed that this result show the same anti-oxidant activity irrespectively time condition. The anti-oxidant activity of ZO-LDH hybrid increased after ultrasound and microwave irradiation in shorter irradiation time. Enhancement activity indicated the ZO release from ZO-LDH hybrid during ultrasound and microwave irradiation. Upon longer irradiation time, anti-oxidant activity of ZO-LDH was maintained with shorter irradiation time which means that really released ZO moieties was protected from ultrasound and microwave irradiation by LDH and LDO. In the manuscript, we revised the figure 6 and table 1 and added discussion as below.
Figure 6. Radical scavenging activity results of (A) ZOE in untreated, ultrasound for 30 minutes and microwave irradiation for 1 minute, ZOE-LDH hybrid untreated and irradiation of (B) ultrasound and (C) microwave
Table 1. IC50 values of radical scavenging of ZOE and ZOE-LDH hybrid
|
IC50 (mg/mL) |
|
ZOE |
ZOE-LDH hybrid |
|
Untreated |
201.81 ± 13.59 |
349.17 ± 47.50 |
Ultrasound for 5 mins |
- |
270.65 ± 28.78a |
Ultrasound for 30 mins |
1003.29 ± 79.82 |
252.46 ± 12.47a |
Microwave irradiation for 1 min |
353.73 ± 29.93 |
259.36 ± 25.42b |
Microwave irradiation for 3 mins |
- |
231.82 ±19.00b |
Page 9, line 269-272: Moreover, upon increasing irradiation tiem, IC50 values of ZOE-LDH revealed 252.46 ± 12.47 ppm for ultrasound (30 mins) and 231.82 ± 19.00 ppm for microwave (3 mins), which was same values with shorter irradiation time as 99% confidence intervals calculated by Student’s t-test
Point 3: The influence of sonication depends on the power of sonicator used. The author should give the power of their sonicator along with a reason why they employed it. It is also not clear why the irradiation time was much longer for the ultrasonication than the microwave irradiation.
Response 3: Natural plant extracts are widely applied in several fields such as food industries, herbal treatment and cosmetics. Using ultrasound in food industry can have several advantages especially in food processing. More effective mixing, faster response, improves heat transfer and less treating time etc. Ultrasound power can be defined as two groups: high energy ultrasound and low energy ultrasound. High energy ultrasound is typically catagorized between 10 and 1000 W/cm2 in intensity and frequencies between 18-100 kHz. (McClements DJ. Advances in the application of ultrasound in food analysis and processing, Trends Food Sci. Technol, 1995;6;293-299) Low energy ultrasound is characterized by high frequencies from 5 MHz to 10MHz, and intensity at lower than 1 W/cm2, normally. (J Chandrapala, Low intensity ultrasound applications on food systems, Int Food Res J, 2015;22;888-895) High energy ultrasound, mainly at range within 20-25 KHz, has been applied for determine the concentration, viscosity and composition of food. (M Gallo et al., Application of Ultrasound in Food Science and technology: A perspective, Foods 2018;7;164) Our ultrasound has frequency at 20 KHz and intensity at 500 W. If anti-oxidant activity of ZO in ZO-LDH can protect from high energy ultrasound, it will also maintain their activity at low energy ultrasound. For this reason, we employed the power of ultrasonic same as above. We exposed extracts at 2.45 GHz microwave radiation. For microwave irradiation time, as the exposing time gets longer, the absorbed power continuously decreases with time. (G Roussy et al., How Microwaves Dehydrate Zeolites, J Phys Chem-US, 1984;88;5702-5708) Based on this research, we set a microwave irradiation time as 1 min and 3 mins.
Point 4: The abbreviation ZO should be defined when first appear in the main text.
Response 4: Thanks for your careful review. In this paper we abbreviated the Zingiber officinale extract as ZO. However, this abbreviation can enough confuse reviewers and readers. Therefore, we revised the abbreviation of Zingiber officinale as ZO and Zingiber officinale extract as ZOE.
Round 2
Reviewer 1 Report
The author correction has significantly improved the manuscript. I recommend its publication.
Reviewer 2 Report
The authors have revised the manuscript following the comments raised. I now recommend publication of this manuscript as is.